# First Record of the Phylum Gnathostomulida in the Southern Ocean

Wolfgang Sterrer [1], Martin V. Sørensen [2], Matteo Cecchetto [3,4,*], Alejandro Martínez [5], Raffaella Sabatino [5], Ester M. Eckert [5], Diego Fontaneto [5] and Stefano Schiaparelli [3,4]

1    Bermuda Natural History Museum, 40 North Shore Road, Hamilton FL04, Bermuda; westerrer@gov.bm
2    Natural History Museum of Denmark, University of Copenhagen, Universitetsparken 15, 2100 Copenhagen, Denmark; mvsorensen@snm.ku.dk
3    Department of Earth, Environmental and Life Sciences (DISTAV), University of Genoa, Corso Europa 26, 16132 Genoa, Italy; stefano.schiaparelli@unige.it
4    Italian National Antarctic Museum (MNA, Section of Genoa), University of Genoa, Viale Benedetto XV No. 5, 16132 Genoa, Italy
5    Molecular Ecology Group (MEG), Water Research Institute (IRSA), National Research Council of Italy (CNR), Largo Tonolli 50, 28922 Verbania, Italy; alejandro.martinezgarcia@cnr.it (A.M.); raffaella.sabatino@irsa.cnr.it (R.S.); estermaria.eckert@cnr.it (E.M.E.); diego.fontaneto@cnr.it (D.F.)
*    Correspondence: matteocecchetto@gmail.com; Tel.: +39-010-3538329

**Abstract:** We report for the first time the occurrence of at least two species of the phylum Gnathostomulida in the Southern Ocean, along the shores of the Ross Sea in Antarctica. At least one species for each of the orders of the phylum (Filospermoidea and Bursovaginoidea) was found using both morphological inspection and DNA metabarcoding of the shallow marine sediments collected with a Van Veen grab or by scuba diving in the area facing the Italian research station "Mario Zucchelli".

**Keywords:** Antarctica; Gnathostomulida; Ross Sea; Ross Sea Region Marine Protected Area; species distribution

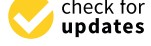



## 1. Introduction

Gnathostomulida Ax, 1956, microscopic worms unique for their monociliated epithelium, were described from detritus-rich, shallow marine sand and are among the most recently discovered animal phyla [1–3]. More than 100 species in two orders (Filospermoidea and Bursovaginoidea) are known to date, mostly from the tropics, but many with a global distribution, from as far north as Disko Island, Greenland and Murmansk, Russia (about 69° N) to Dunedin, New Zealand (about 46° S). Knowledge of this phylum is very limited, and their occurrence is often overlooked in studies of marine microscopic animals [4].

The aim of the current note is to provide unambiguous evidence of the previously unnoticed occurrence of Gnathostomulida as far south as the shores of Antarctica in the Southern Ocean. This record represents the highest absolute latitude reached by the phylum, which is now reported down to 74° S.

## 2. Materials and Methods

Sampling activities were carried out during the Austral summer 2018–2019 in the framework of the Italian National Antarctic Program (PNRA) project "TNB-CODE" (Terra Nova Bay barCODing and mEtabarcoding of Antarctic organisms from marine and limno-terrestrial environments). Sediment samples were collected either with a van Veen grab deployed from a dinghy or by SCUBA divers in the area nearby the Italian Research Station "Mario Zucchelli" along the Terra Nova Bay shores (Ross Sea) (Figure 1, Table 1). Different types of substrates were targeted for sampling (e.g., gravel, sand, silt, spicule mat, etc.), based on available sedimentological maps [5] and previous knowledge, at depths

ranging from the intertidal down to 70 m. Sediment samples were not meant to provide a quantitative estimate of abundances, but only to provide a preliminary survey of marine meiofauna in the area. Samples were temporarily preserved in plastic buckets or jars and immediately brought back to the laboratory in the "Mario Zucchelli" research station. Part of the sample was inspected visually under a stereomicroscope to sort meiofauna, take photos of the living organisms, and store them in ethanol as vouchers. Another part of the sample was passed through sieves of mesh size from 500 to 20 μm, with the organisms passing the filtering at 500 μm but not the 20 μm stored in ethanol at −20 °C for DNA metabarcoding. Mesh sizes were selected to collect all organisms belonging to the ecological guild of microscopic animals of the meiofauna [6]. All samples, both for morphology and for DNA metabarcoding, were processed with a preliminary extraction using magnesium chloride, as it is common for soft-bodied meiofauna [7].

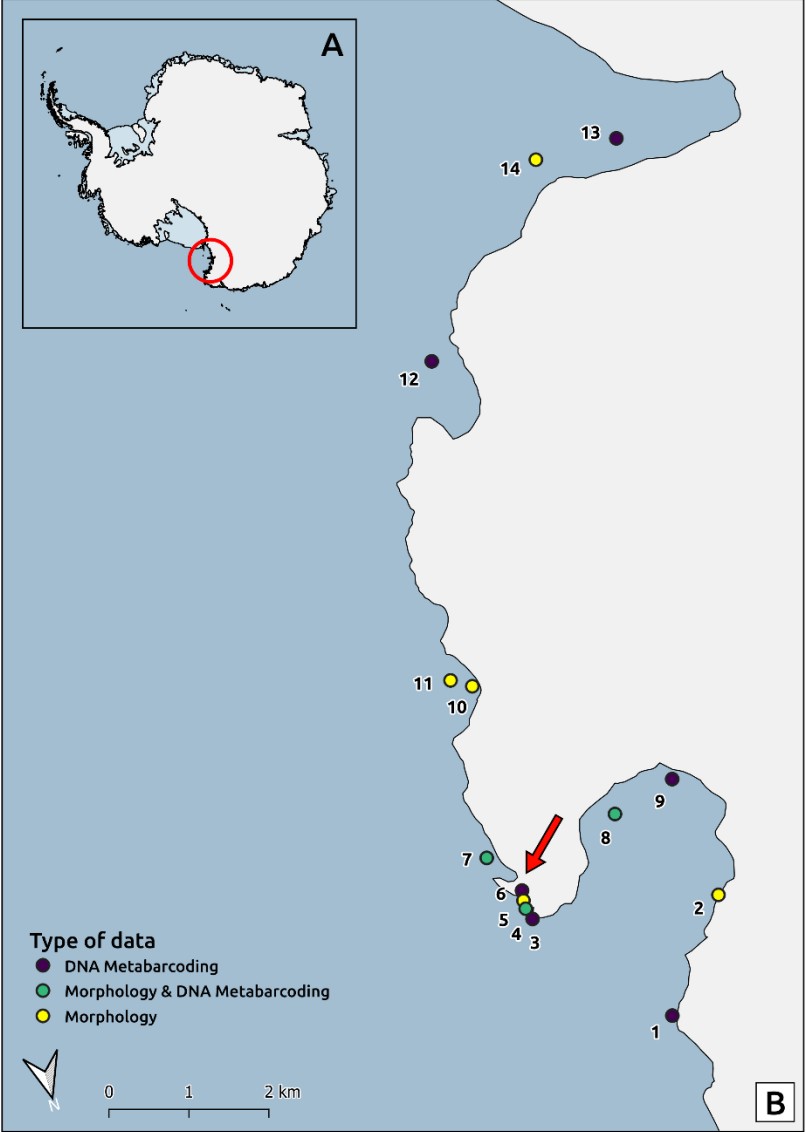

**Figure 1.** Location of Terra Nova Bay (Ross Sea) in Antarctica (**A**) and study area with sampled stations (**B**). The red arrow indicates the position of Mario Zucchelli research station. Sampling sites that appear to lie on the coastline represent very shallow, near shore sites. Numbers refer to sampling stations listed in Table 1 (field "STATION-ID"). This map was produced using the collection of datasets "Quantarctica" [8] and QGIS [9].



**Table 1.** Number of Gnathostomulida specimens found from morphological inspection and number of reads from DNA metabarcoding ("MORPHOLOGY" and "DNA METABARCODING" fields, respectively) of marine sediment samples collected around the Italian Research Station "Mario Zucchelli" in the Ross Sea in Antarctica. The table includes fields referring to the museum voucher codes of the specimens inspected by morphological examination and preserved at the Italian National Antarctic Museum ("MNA#"), the sampling stations' labels as reported in Figure 1 ("STATION-ID"), the name of the sampling station ("EVENT-ID"), type of habitat ("HABITAT"), sampling gear or methodology adopted ("GEAR"), depth (m), sampling date and geographic coordinates in WGS84.

| MNA# | STATION-ID | EVENT-ID | MORPHOLOGY | DNA METABAR-CODING | HABITAT | GEAR | DEPTH (m) | DATE | LATITUDE | LONGITUDE |
|---|---|---|---|---|---|---|---|---|---|---|
| | 1 | PN 2 | - | 1 | sediment, silty gravel | diving | 24 | 22 December 2018 | −74.6756 | 164.0695 |
| | 1 | PN 4 | - | 15 | sediment, silty gravel | diving | 24 | 22 December 2018 | −74.6756 | 164.0695 |
| | 1 | PN 5 | - | 2 | sediment, silty gravel | diving | 24 | 22 December 2018 | −74.6756 | 164.0695 |
| MNA-14743 | 2 | AMORPHOUS 2 | 1 | - | sediment, fine sand | diving | 20 | 28 December 2018 | −74.6872 | 164.0366 |
| MNA-14061, MNA-14744 | 2 | AMORPHOUS 3 | 2 | - | sediment, fine sand | diving | 20 | 28 December 2018 | −74.6872 | 164.0366 |
| | 3 | MPN 1 | - | 2 | epilithic, bryozoans | diving | | 21 January 2019 | −74.6903 | 164.1152 |
| | 3 | MPN 4 | - | 1 | sediment, silty sand | diving | | 21 January 2019 | −74.6903 | 164.1152 |
| MNA-14069, MNA-14070 | 4 | MAREOGR 2 | 3 | - | sediment, sand coarse | grab | 49 | 13 January 2019 | −74.6917 | 164.1168 |
| | 4 | MAREOGR 1 | - | 4 | sediment, silty sand | grab | 25 | 26 December 2018 | −74.6917 | 164.1168 |
| MNA-14071 to MNA-14079 | 5 | MOL 1 | 20 | - | sediment, gravel | grab | 20 | 13 January 2019 | −74.6926 | 164.1168 |
| | 6 | Spiaggia Molo 1 | - | 2 | sand coarse | hand | 0 | 26 December 2018 | −74.6937 | 164.1162 |
| MNA-14080 | 7 | ADA 1-4 | 1 | 14 | sediment, silt | grab | 70 | 17 January 2019 | −74.6983 | 164.1268 |
| MNA-14746 | 7 | ADA 1-5 | 1 | 1 | gravel | grab | 19 | 17 January 2019 | −74.6983 | 164.1268 |
| | 7 | ADA 1-3 | - | 30 | sediment, silty sand | grab | 40 | 30 December 2018 | −74.6983 | 164.1268 |

**Table 1.** *Cont.*

| MNA# | STATION-ID | EVENT-ID | MORPHOLOGY | DNA METABAR-CODING | HABITAT | GEAR | DEPTH (m) | DATE | LATITUDE | LONGITUDE |
|---|---|---|---|---|---|---|---|---|---|---|
| MNA-14085 | 8 | TETH 5 | 1 | 612 | spicule mat | grab | 70 | 30 January 2019 | −74.6991 | 164.0694 |
| | 8 | TETH 6 | - | 9759 | spicule mat | grab | 70 | 31 January 2019 | −74.6991 | 164.0694 |
| | 9 | Spiaggia 1 | - | 2 | sediment, silt | diving | 22 | 19 December 2018 | −74.7011 | 164.042 |
| | 9 | Spiaggia 2 | - | 216 | sediment, silt | diving | 22 | 19 December 2018 | −74.7011 | 164.042 |
| | 9 | Spiaggia 5 | - | 16 | sediment, silt | diving | 22 | 19 December 2018 | −74.7011 | 164.042 |
| MNA-14062 to MNA-14066 and MNA 14745 | 10 | FAR 2 | 6 | - | sediment, sand | grab | 19 | 5 January 2019 | −74.7173 | 164.1128 |
| MNA-14067, MNA-14068 | 11 | FAR 1 | 2 | - | sediment, silty sand | grab | 40 | 5 January 2019 | −74.7186 | 164.121 |
| | 12 | CAL1 | - | 28 | sediment, silt | grab | 50 | 27 January 2019 | −74.7536 | 164.0915 |
| | 13 | ADCO 1 | - | 8 | epilithic, bryozoans | grab | 33 | 3 January 2019 | −74.7719 | 163.9897 |
| | 13 | ADCO 2 | - | 4 | sediment, silt | grab | 33 | 3 January 2019 | −74.7719 | 163.9897 |
| MNA-14081 to MNA-14084 and MNA-14747 | 14 | ADCO 8 | 10 | - | sediment, sand | grab | 20 | 27 January 2019 | −74.7721 | 164.0252 |

DNA was extracted from ethanol-stored samples using the commercial PowerSoil extraction kit (Qiagen). The primers used for Illumina sequencing targeted a DNA fragment of approximately 450 base pairs corresponding to the V1–V2 regions of the nuclear small subunit rRNA gene (18S rDNA). They were based on the commonly used 18SF04 (5′-GCTTGTCTCAAAGATTAAGCC-3′) for V1-V2 region of 18S [10–12] and a modified version of 18SR22, called 18SR22 (5′-GCCTGCTGCCTTCCTTRGA-3′) as in Martínez [13]. Sequencing was performed on the NovaSeq Illumina platform using the 2 × 250 paired-end (PE) approach. The pipeline to obtain unique sequences called amplicon sequence variants (ASV) was that of Martínez [13], including a maximum number of nucleotides allowed to differ in the merging operation set to 10 and a minimum length of merged sequences corresponding to 250 bp following the UPARSE pipeline [14]. Adapters and primer removal was previously performed using cutadapt [15], whereas the clustering procedure was conducted using the UNOISE algorithm implemented in USEARCH [16]. The sequences that passed the bioinformatics quality filtering steps were blasted in GenBank for an initial identification to the phylum level, and all sequences that had a potential hit to the phylum Gnathostomulida were retained. The retained sequences (Supplementary File S1) were then aligned with a reference set of clean gnathostomulid 18S sequences downloaded from GenBank. The aligned dataset, trimmed to the length of the DNA sequences from metabarcoding, was used to obtain a visual representation of the phylogenetic relationships through a neighbor-joining tree, from the R v4.1.3 [17] package ape v5.6.2 [18].

All gnathostomulid samples are part of the collections of the Italian National Antarctic Museum (MNA, Section of Genoa, voucher codes MNA-14061 to MNA-14085 and MNA-14743 to MNA-14747).

## 3. Results

The two analyses conducted in parallel, i.e., morphological inspection and DNA metabarcoding, both supported the presence of Gnathostomulida in the analyzed samples. The morphological inspection allowed isolating a total of 47 individual organisms belonging to the phylum Gnathostomulida obtained from 10 sampling stations (Table 1) by visually sorting the collected sediment. Organisms of both orders, Filospermoidea and Bursovaginoidea, were found (Figure 2). Similarly, the DNA metabarcoding analyses supported the occurrence of organisms belonging to two separate clades, one in each order of Gnathostomulida (Figure 3), by producing 10,717 sequences, which clustered in 10 unique sequences (ASV) from 18 sampling events (Table 1). Interestingly, in only three sampling sites, the occurrence of the phylum was supported both from morphological inspection and DNA metabarcoding (Table 1).

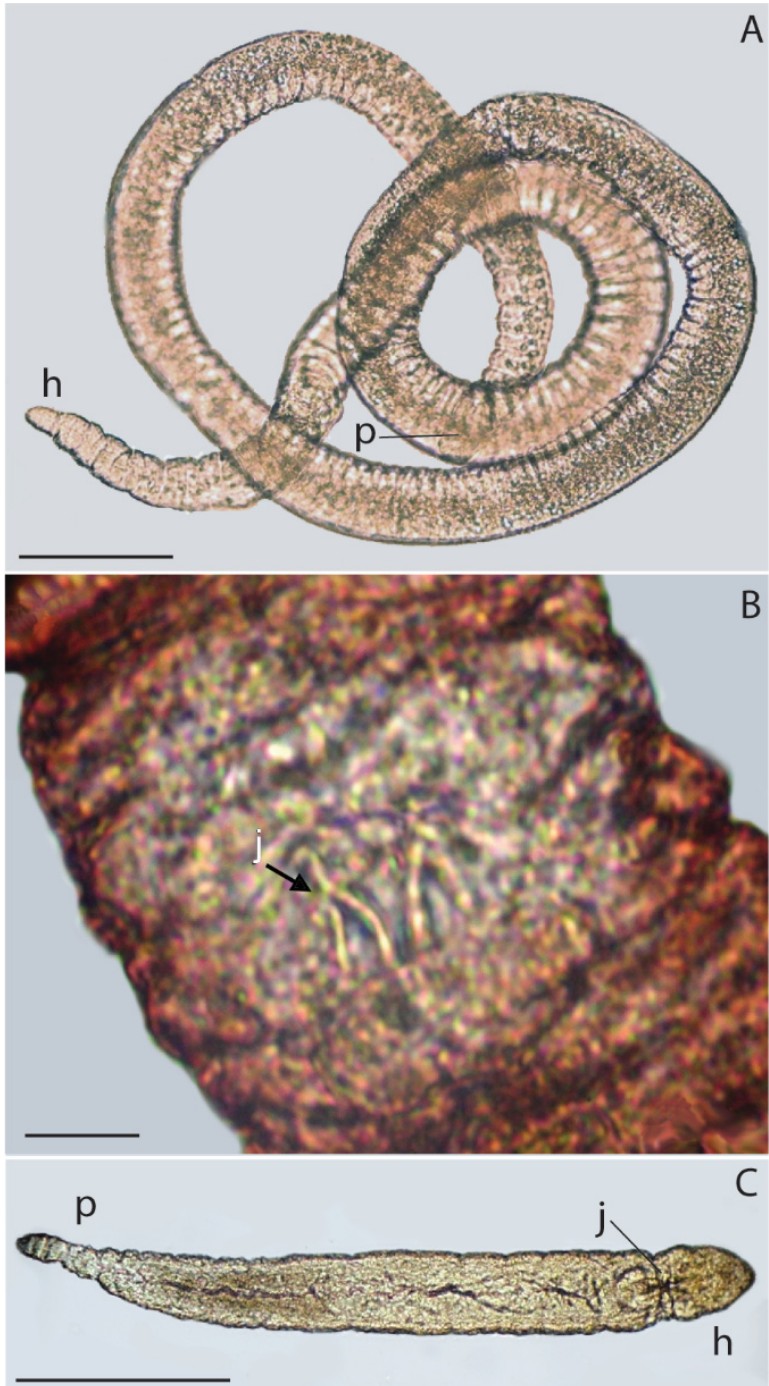

**Figure 2.** Light micrographs with live specimens of Gnathostomulida from Antarctica. (**A**) A filospermoid, with a pointed anterior and a rounded posterior. (**B**) Jaws of a red filospermoid, *Haplognathia* cf. *ruberrima* (Sterrer, 1966). (**C**) A juvenile bursovaginoid (Austrognathiidae), with rounded anterior and tailed posterior. Abbreviations: head (h), jaws (j), posterior end (p). Scale bars: (**A**) = 100 μm, (**B**) = 10 μm, (**C**) = 200 μm.

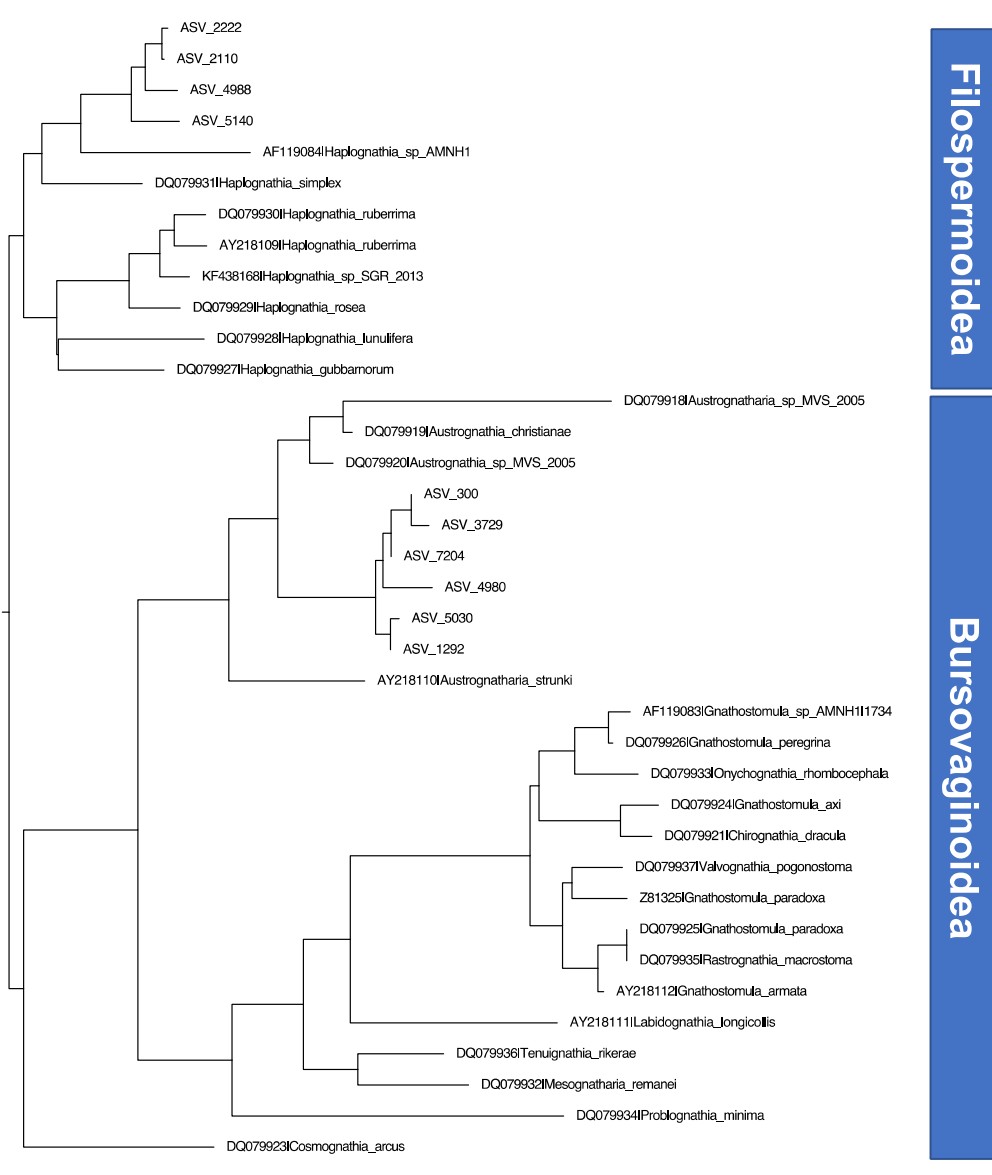

**Figure 3.** Neighbor-joining tree of the 10 unique sequences found from DNA metabarcoding of the analyzed samples (named as ASV followed by an identification number) and the reference library of 18S sequences from each available species, obtained from GenBank (named as accession number followed by species name).

## 4. Discussion

This is the first record of the phylum from the Southern Ocean along the shores of Antarctica, and it represents the most extreme latitudinal record for both hemispheres by reaching 74° S. Although the collected material has yet to be analyzed in detail, morphology and DNA metabarcoding concur in supporting that the samples from Antarctica contain at least one species of each order, expanding the geographic coverage of the phylum to also include the only continent where Gnathostomulida were not known to occur.

The type of substrate where gnathostomulids were found ranges from silt to sand and gravel but also includes spicule mats and epilithic biofilms with encrusting bryozoans (Table 1). Such habitats represent the usual ones where organisms of the phylum are known to live (i.e., sand with detritus), but finding them in spicule mats has not been reported previously [3,4].

We have to acknowledge that some of the samples where only one or few sequences were found from DNA metabarcoding could represent instances of contamination [19]. However, samples wherein hundreds of sequences were found should be considered reliable.

The discrepancy between the detection of gnathostomulids based on visual inspection and based on DNA metabarcoding underlines the difficulties of comparing results from different approaches. On the one hand, it is surely difficult to retrieve live specimens of meiofauna from samples in Antarctica: it is highly likely that for some samples, the visual inspection was not carefully performed, making it possible that animals were not seen, even if present. On the other hand, DNA metabarcoding is known to produce false positives, especially in the samples with very few reads: in our datasets, the sites where only one or two reads were found may represent false positives (Table 1). Sites such as TETH 6, with thousands of reads, may be considered reliable.

**Supplementary Materials:** The following supporting information can be downloaded at: https://www.mdpi.com/article/10.3390/d14050382/s1, Supplementary File S1: Alignment of the sequences used to obtain the tree in Figure 3, in fasta format.

**Author Contributions:** Conceptualization, D.F. and S.S.; methodology, W.S., M.V.S., M.C., A.M., R.S., E.M.E., D.F. and S.S.; formal analysis, W.S., M.V.S., M.C., A.M., R.S., E.M.E., D.F. and S.S.; data acquisition M.C., A.M., R.S. and E.M.E.; data curation, M.C., A.M., R.S. and E.M.E.; writing—original draft preparation, W.S., M.V.S., D.F. and S.S.; writing—review and editing, M.C., A.M., R.S. and E.M.E.; funding acquisition, S.S. All authors have read and agreed to the published version of the manuscript.

**Funding:** Sampling activities were performed within the PNRA project "TNB-CODE" (Terra Nova Bay barCODing and mEtabarcoding of Antarctic organisms from marine and limno-terrestrial environments) (PNRA 16_00120, PI: Stefano Schiaparelli). Authors are grateful to the Italian National Antarctic Scientific Commission (CSNA) and the Italian National Antarctic program (PNRA) for the endorsement of the Special Issue initiative and to the Italian National Antarctic Museum (MNA) for the financial support.

**Institutional Review Board Statement:** Not applicable.

**Data Availability Statement:** In accordance with FAIR principles, the DNA sequence dataset (Supplementary File S1) is available in the Supplemental Materials.

**Acknowledgments:** We are indebted with Ellenico Francesco and Gainnoni Paolo (Gruppo Operativo Subacquei COMSUBIN, Marina Militare Italiana) and Corti Ramon (Gruppo Operativo Incursori, COMSUBIN, Marina Militare Italiana) for logistic support during diving operations and sampling at sea. This paper is an Italian contribution to the Commission for the Conservation of Antarctic Marine Living Resources CONSERVATION MEASURE 91-05 (2016) for the Ross Sea Region Marine Protected Area, specifically, addressing the priorities of Annex 91-05/C.

**Conflicts of Interest:** The authors declare no conflict of interest. The funders had no role in the design of the study; in the collection, analyses, or interpretation of data; in the writing of the manuscript; and in the decision to publish the results.

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
