# Peer review of "First Record of the Phylum Gnathostomulida in the Southern Ocean"

_diversity, doi:10.3390/d14050382_

Round 1

Reviewer 1 Report

Review of a paper by Wolfgang Sterrer, Martin V. Sørensen, Matteo Cecchetto, Alejandro Martínez, Raffaella Sabatino, Ester M. Eckert, Diego Fontaneto  and Stefano Schiaparelli entitled First record of the phylum Gnathostomulida in the Southern  Ocean.

The paper is devoted to the first discovery of Gnathostomulida representatives in the coastal waters of Antarctica. Specifically, along the coast of Victoria Land.

As a zoological publication, it seems to be very unusual, since the morphological description and illustrations of the animals found are not directly related to those samples that were studied by the metabarcoding method.

However, the use of such an unusual methodology is justified by the really the very interesting zoological findings and the enormous technical difficulties that meiobenthic researchers face in field work on the coast of Antarctica. I think the paper could be accepted for publication with minor improvements.

My comments:

  1. In Figure 1, it is necessary to place a scale bar so that the reader can understand what part of the coast was surveyed.
  2. The authors very briefly describe molecular genetic procedures, but this, in principle, is correct, especially since they indicate the sequences of the primers used, and this, in general, is the main thing. But, since they are based on sequences obtained on the Illumina, they must specify, what type of the Illumina platform have been used? Since different types of platforms give a different percentage of errors.
  3. Authors should describe the filtering process more a bit more detailed, since the "comb" on the tree can be due to both genetic diversity and a low filtering threshold, leading to "diversity generation" due to random Illumina errors.

Author Response

Response to Reviewer 1 Comments

Point 1: In Figure 1, it is necessary to place a scale bar so that the reader can understand what part of the coast was surveyed.

Response 1: A scale bar depicting a 2 km distance was added to the figure as suggested by the reviewer.

Point 2: The authors very briefly describe molecular genetic procedures, but this, in principle, is correct, especially since they indicate the sequences of the primers used, and this, in general, is the main thing. But, since they are based on sequences obtained on the Illumina, they must specify, what type of the Illumina platform have been used? Since different types of platforms give a different percentage of errors.

Response 2: The sequencing platform was specified in the Materials and Methods section by including an additional sentence, after the primers description (now lines 73-74): “Sequencing was performed on the NovaSeq Illumina platform using the 2 × 250 paired-end (PE) approach.”

Point 3: Authors should describe the filtering process more a bit more detailed, since the "comb" on the tree can be due to both genetic diversity and a low filtering threshold, leading to "diversity generation" due to random Illumina errors.

Response 2: The description of the pipeline was expanded, after the sentence on the sequencing platform referring to the previous comment, including the most important bioinformatics quality filtering steps and clustering process (now lines 74-79).

Reviewer 2 Report

This short article reports on the occurrence of gnathostomulids along the coastal margin of the Ross Sea. Gnathostomulids are a neglected component of meiobenthic communities — yet, they can be extremely abundant in certain types of marine substrates. This manuscript is the first to report representatives of this phylum from Antarctica. This fact alone is highly significant and makes this paper an excellent contribution to this special issue on the biodiversity of the Ross Sea. As such, I would be delighted to see this short article published.

I only have a few minor comments/suggestions, but other than that I think the article can be published in its current form.

Lines 58–59: “Part of the sample was inspected visually …” It is not entirely clear if any processing of the substrate was done before this. Extraction methods for soft-bodied meiofauna such as ‘magnesium chloride’ or ‘sea ice’ are commonly used to separate meiofauna from the sediment/substrate before isolating individual specimens under the stereoscope. However, here it seems like the actual substrate was inspected without such pre-processing, which would be extremely time-consuming? Maybe this could be clarified here.

Lines 128–131: This is just a suggestion, but I think this last paragraph could be a bit more streamlined+expanded. For instance: "The discrepancy between the detection of gnathostomulids based on visual inspection and based on DNA metabarcoding underlines the difficulties of retrieving live specimens from samples in Antarctica." Then I would list some potential reasons why visual inspection of the samples could sometimes be problematic. It would also be nice to add some recommendations to remediate this in the future.

Figure 2. Maybe some of the images could be annotated with abbreviations for the visible structures? For the uninitiated it might be difficult to know what the actual jaw is in Fig. 2B. In addition, you can clearly distinguish the rostral and caudal apophyses. In Fig. 2C, you can also see the outline of the pharynx.

I do not require anonymity. Signed: Niels Van Steenkiste

Author Response

Response to Reviewer 2 Comments

Point 1: Lines 58–59: “Part of the sample was inspected visually …” It is not entirely clear if any processing of the substrate was done before this. Extraction methods for soft-bodied meiofauna such as ‘magnesium chloride’ or ‘sea ice’ are commonly used to separate meiofauna from the sediment/substrate before isolating individual specimens under the stereoscope. However, here it seems like the actual substrate was inspected without such pre-processing, which would be extremely time-consuming? Maybe this could be clarified here.

Response 1: A new sentence (now lines 64-66) was included at the end of the paragraph explaining the pre-processing adopted to separate meiofauna from the sediment, including the citation to Curini-Galletti et al. (2012): “All samples, both for morphology and for DNA metabarcoding, were processed with a preliminary extraction using magnesium chloride, as it is common for soft-bodied meiofauna.”

Point 2: Lines 128–131: This is just a suggestion, but I think this last paragraph could be a bit more streamlined+expanded. For instance: "The discrepancy between the detection of gnathostomulids based on visual inspection and based on DNA metabarcoding underlines the difficulties of retrieving live specimens from samples in Antarctica." Then I would list some potential reasons why visual inspection of the samples could sometimes be problematic. It would also be nice to add some recommendations to remediate this in the future.

Response 2: The suggestion was appreciated and the paragraph has been expanded and modified in order to properly describe the mentioned issue (now lines 144-152):

“The discrepancy between the detection of gnathostomulids based on visual inspection and based on DNA metabarcoding underlines the difficulties of comparing results from different approaches. One the one hand, it is surely difficult to retrieve live specimens of meiofauna from samples in Antarctica: it is highly likely that for some samples the vis-ual inspection was not carefully performed, making likely that animals were not seen, even if present. On the other hand, DNA metabarcoding is known to produce false posi-tives, especially in the samples with very few reads: in our datasets, the sites where only one or two reads were found may represent false positives (Table 1). Sites like TETH 6, with thousands of reads, may be considered reliable.”

Point 3: Figure 2. Maybe some of the images could be annotated with abbreviations for the visible structures? For the uninitiated it might be difficult to know what the actual jaw is in Fig. 2B. In addition, you can clearly distinguish the rostral and caudal apophyses. In Fig. 2C, you can also see the outline of the pharynx.

Response 2: The image has been modified in order to include additional specifications on the morphological characters depicted. The figure caption has been changed accordingly.

Reviewer 3 Report

Dear editor and authors of the manuscript “First record of the phylum Gnathostomulida in the Southern Ocean”.

I have now finished my review of your manuscript submitted to “Diversity”.

The manuscript is a useful addition to the gnathostomulida literature and presents the data clearly.

However, I have three main concerns to point out:

i) in table 1, it was a little trick to understanding the meaning of MNA and MAP-ID and I think that it could be explained in more detail in the captions of the table. Another difficult that I had was to link the sampled stations -fig. 1-  mentioned in the captions with the information in table 1;

ii) In the results, I missed more details about the species or morphotypes (e.g. names, number of specimens) that were found using morphological analysis. The authors said that they found specimens belonging to the both order, but without any further details;

iii) again it was little trick to understand the figure 2, because I did not understand what “ASVs” means in the present study. I got that ASVs were sequences obtained in present analysis, but I could not understand which are these specimens, as I could not find more information in the results (I commented about it in “ii”). Another suggestion in figure 2, the authors could, at least, indicate which branch is Filospermoidea or Bursovaginoidea.

Author Response

Response to Reviewer 3 Comments

Point 1: i) in table 1, it was a little trick to understanding the meaning of MNA and MAP-ID and I think that it could be explained in more detail in the captions of the table. Another difficult that I had was to link the sampled stations -fig. 1-  mentioned in the captions with the information in table 1;

Response 1: The caption of Table 1 has been modified in order to provide a better explanation of the different fields included. In order to avoid confusion, the field “MAP-ID” has been changed to “STATION-ID”, whereas “STATION” has been changed to “EVENT-ID”, as these names explain more properly the actual meaning of the field. The caption of Figure 1 has been changed accordingly.

Point 2: ii) In the results, I missed more details about the species or morphotypes (e.g. names, number of specimens) that were found using morphological analysis. The authors said that they found specimens belonging to the both order, but without any further details;

Response 2: The number of specimens for each sampling station is reported in Table 1, for which a more detailed description has been included following the previous suggestion, in order to clarify the meaning of each field. The scientific names for the specimens analyzed are not included as not all of those specimens were identified to species level. For this reason, only the total specimen number for each group of museum voucher codes is reported. Some records for the “MNA#” field in Table 1 were adjusted as were wrongly reported as ranges (e.g. MNA-14061 to MNA-14744) instead of two specific different voucher codes. The example reported above is now changed to “MNA-14061, MNA-14744” in the correct version of Table 1.

Point 3: iii) again it was little trick to understand the figure 2, because I did not understand what “ASVs” means in the present study. I got that ASVs were sequences obtained in present analysis, but I could not understand which are these specimens, as I could not find more information in the results (I commented about it in “ii”). Another suggestion in figure 2, the authors could, at least, indicate which branch is Filospermoidea or Bursovaginoidea.

Response 2: The sequences reported as “ASVs” refer to amplicon sequence variants obtained from the DNA metabarcoding technique described in the Materials and Methods section. For this reason, the specimens reported in Table 1, and referred to in the Results, are only those that have been sorted and analyzed through morphological inspection, whereas the ASVs come from the “parallel” analyses conducted on the same starting material, but for which no specimen has been sorted. In order to avoid confusion, a better explanation of the obtained results has been included in the Results paragraph (lines 92-102), clearly discriminating the two different kind of outputs from the analyses conducted. Figure 3 has been changed according to the comment by including two bars that allow the reader to easily identify the two different orders to which each sequence belongs to.